# Otogenic Meningitis or Meningoencephalitis in 30 Dogs: Association Between Neurological Signs, Magnetic Resonance Imaging Findings, and Outcome

**DOI:** 10.3390/vetsci12050456

**Published:** 2025-05-09

**Authors:** Meritxell Moral, Carlos Blanco, Valentina Lorenzo

**Affiliations:** Clínica Neurología Veterinaria, C/Diseño, 26, 28906 Getafe, Madrid, Spain

**Keywords:** otitis media-interna, bacterial meningoencephalitis, magnetic resonance imaging, CSF, neurology, dog

## Abstract

This retrospective study aimed to assess the accuracy of neurological examination in identifying intracranial extension of ear infection in dogs and to investigate the clinical data that may potentially aid in its diagnosis and prognosis. The neurological examination was accurate in localizing the lesion in the central nervous system in 33% of dogs. This study supports the conclusions that otogenic meningoencephalitis should be considered even in patients with peripheral vestibular syndrome (PVS), and that magnetic resonance imaging (MRI) and cerebrospinal fluid (CSF) should be advised for diagnosis in certain cases. Although it bears grave consequences for the patient if not recognized and addressed appropriately, excellent outcomes can be achieved with medical management. Further studies, which include comprehensive ancillary diagnostics with cultures and surgical treatments, are needed to identify the most common pathogens and to elaborate treatment guidelines.

## 1. Introduction

Bacterial meningitis or meningoencephalitis due to an intracranial extension of otitis media-interna (OMI) is an uncommon but severe condition requiring urgent therapeutic interventions [1,2]. This pathology has been described for several decades in human medicine, but in dogs and cats, the information is limited to a few studies [1,2,3] and case reports [4,5,6,7,8,9,10].

On neurological examination, OMI may induce a peripheral vestibular syndrome (PVS) characterized by vestibular ataxia and cranial nerve VIII deficits with normal proprioception, sometimes with cranial nerve VII or sympathetic nerve deficits. In the case of intracranial extension of the infection, proprioceptive deficits, abnormal mental status, and other central nervous system (CNS) signs may arise. Nevertheless, the neurological examination by itself is not always efficient in identifying intracranial lesions in vestibular disease [11].

Magnetic resonance imaging (MRI) is the gold standard technique to detect lesions in the middle and inner ear and surrounding soft tissues and to confirm the extension of OMI to the CNS [1]. MRI also allows for obtaining high resolution three-dimensional (3D) images [12,13].

This study aimed to determine the diagnostic accuracy of neurological examination in identifying intracranial extension in canine otitis media-interna, and to evaluate the associations between MRI findings, clinical variables, and patient outcomes.

## 2. Materials and Methods

### 2.1. Study Population and Inclusion Criteria

The case records of dogs presented to a referral neurology center from June 2015 to August 2023 were reviewed. The inclusion criteria were dogs with clinical signs of vestibular dysfunction (peripheral, or central with or without other CNS signs), MRI of the head that documented soft tissue material within the tympanic bulla/bullae with contrast enhancement of surrounding soft tissues with or without evidence of extension to the adjacent meninges or neuroparenchyma, no medical history of previous intracranial CNS disease, neutrophilic pleocytosis (>50%) with degenerate neutrophils in the cerebrospinal fluid (CSF) analysis if performed, and recorded short-term (from the diagnosis to 6 weeks) and long-term outcome (from 6 weeks to a minimum of 12 weeks following diagnosis).

Data retrieved from the medical records included signalment (age, weight, sex, breed), history of past or current otitis externa (OE), physical examination, blood cell count, serum biochemistry, and treatments (before admission and after diagnosis). If performed, results of bacterial culture from urine, blood, CSF and/or middle ear samples were also noted.

### 2.2. Neurological Examination

Neurological examinations were performed by either a resident in training or a Diplomate of the European College of Veterinary Neurology (ECVN). Dogs were classified as having a PVS or a central vestibular syndrome (CVS) (either alone or together with other intracranial signs). The onset of neurological signs was classified as acute (≤48 h), subacute (3–7 days) or chronic (>7 days). Vestibular ataxia was classified as mild (able to walk with mild to moderate imbalance), moderate (difficulty in walking, spontaneous falls), or severe (unable to stand or walk).

### 2.3. Imaging Protocol Magnetic Resonance Imaging (MRI) and Classification

All dogs were anesthetized and MRI studies of the head were acquired with 1.5 Tesla MRI (Gyroscan Intera, and Prodiva CS, Philips, Eindhoven, The Netherlands). All imaging series comprised a minimum of T1-weighted (T1W), T2-weighted (T2W), and T2W fluid attenuation inversion recovery (FLAIR) sequences and included transverse, sagittal, and dorsal images, with T1W images acquired before and after intravenous (IV) administration of gadolinium contrast agent (Dotarem^®^ 0.5 mmol/mL Guerbet, Gadoteric acid). High-resolution 3D T1W sequences were included in some studies to better evaluate the presence of facial or vestibulocochlear nerve changes. All imaging studies were reviewed independently for diagnostic accuracy by a board-certified neurologist and a neurology resident in training.

MRI images were evaluated for the presence of inflammatory changes which included: soft tissue material within one or both tympanic cavities (T2W hyperintense with T1W post-contrast enhancement), changes in para-aural soft tissues (diffuse hyperintensity in T2W and isointensity in T1W with post-contrast enhancement), meninges (T1W post-contrast enhancement), brain parenchyma (T2W hyperintensities, or suspected abscess with a T1W-hypointense, T2W-hyperintense center, and strongly-contrast enhancing rim), and the accumulation of extra-axial material corresponding to the formation of empyema (T2W-hyperintense, T1W-hypointense with peripheral contrast enhancement) [3]. The degree of contrast enhancement was described as mild, moderate, or marked, and the pattern as homogeneous or heterogeneous. The presence of nerve (facial and/or vestibulocochlear) thickening and/or enhancement was also assessed.

Signs of increased intracranial pressure (ICP) were also considered (mass effect with parenchymal shift with or without brain herniation, perilesional edema, and effacement of the cerebral sulci) [14].

The extension and location of the lesions were classified as follows:-Group A: bulla/bullae occupation and changes in surrounding soft tissues without evidence of extension to the adjacent meninges or neuroparenchyma.-Group B: bulla/bullae occupation, changes in surrounding soft tissues and imaging of meningeal post-contrast enhancement.-Group C: bulla/bullae occupation, changes in surrounding tissues, meningeal enhancement, and brainstem lesion with or without the formation of an empyema (Figure 1).

If performed, follow-up MRI results were studied. Good radiological improvement was considered if the tympanic cavities occupation was reduced by 25% or greater, there was no soft tissue enhancement, and/or no signs of abscess or empyema if it was previously present.

### 2.4. Cerebrospinal Fluid (CSF) Analysis

CSF analysis was collected from the cerebellomedullary cistern (if no signs of foramen magnum or rostral transtentorial parenchymal herniation were observed in MRI) and was evaluated for cellular content and cytology. Total nucleated cell count (TNCC) of <5 cells/µL and CSF microprotein concentration of <25 mg/dL, were considered within the reference interval. Neutrophilic pleocytosis was considered if >50% were neutrophils. If performed, culture and follow-up CSF results were noted. Diagnosis of bacterial meningitis or meningoencephalitis was considered to include (1) the presence of intracellular bacteria on CSF cytological evaluation or (2) positive CSF bacteriological culture or (3) neutrophilic pleocytosis with degenerate neutrophils on CSF cytological evaluation with clinical improvement after initiation of administration of antibiotics, or these findings in combination [3].

### 2.5. Follow-Up

The information about the short-term outcome (from diagnosis up to the first 6 weeks) was obtained through clinical evaluations conducted by a neurology resident. The long-term outcome (up to 6 months) was obtained by telephone with the owner, referring veterinarian, or both. The median recovery time was considered from the start of medication to a normal neurological examination or until clinical improvement plateaued. The persistent neurological deficits associated with the condition were recorded based on the neurological examination and owners’ perception and were considered sequels.

### 2.6. Statistical Analysis

To study the relationship between the MRI results and the neuro-anatomical localization, as well as the presence of sequels, the MRI group variable was converted to binary, where those cases belonging to group A (no signs of intracranial extension) were assigned a value of 0, and those in B or C (signs of intracranial extension) were 1, and Ordinary Least Squares (OLSs) regressions were performed.

The possible relationship between the MRI results and specific clinical variables (age, previous history of chronic or recurrent otitis, external otitis on admission, use of antibiotic, glucocorticoid (GCC) or non-steroid anti-inflammatory drugs (NSAIDs) before diagnosis, grade of vestibular ataxia on admission, and presence of leucocytosis in blood) was also evaluated, as well as the relationship of the presence of sequels with other variables (age, previous history of chronic or recurrent otitis, external otitis on admission, use of antibiotic, GCC or NSAIDs before diagnosis, grade of vestibular ataxia on admission, type and duration of antibiotic treatment, and duration of glucocorticoid treatment if administered). All analyses were performed with Stata 17, and *p* values < 0.05 were considered statistically significant.

In addition, the correlation between the neuro-anatomical localization on admission and the MRI grading (A, B, C) was assessed. Regarding the outcome, the correlation between the presence of sequels and the MRI results, as well as the type and duration of treatment was studied.

## 3. Results

A table with full details is provided in the Appendix A.

### 3.1. Signalment

A total of 30 dogs fulfilled the inclusion criteria. The median age was 7 years (range 1.5–14 years) and the median weight was 13.6 kg (range 1.2–27.7 kg), comprising 12 sprayed females, 11 neutered males, and 7 entire males. The represented breeds included French bulldogs (n = 24), English bulldogs (n = 4), Cavalier King Charles Spaniel (n = 1), and Yorkshire Terrier (n =1), with 29/30 (96%) corresponding to brachycephalic breeds.

### 3.2. History, Physical Examination, and Ancillary Studies

A history of chronic or recurrent OE was reported in 14/30 (47%) dogs and included all dogs from MRI group C (7/7), 33% of MRI group A (2/6), and 29% of MRI group B (5/17).

OE at the time of admission was detected in 10/30 (33%) of dogs, including 5/10 of MRI group B, 4/10 dogs of MRI group C, and 1/10 of MRI group A. In 3/30, a superficial unilateral corneal ulcer was present, and 8/30 showed moderate signs of pain, including cervical (1/8) or aural (5/8) pain, or discomfort on opening the mouth (2/8).

The reported treatment before the admission included an antibiotic (median time 4 days) on 11/30: cephalexin (n = 4), amoxicillin-clavulanic acid (ACA) (n = 2), ciprofloxacin (n = 1), cefovecin (n = 1), enrofloxacin (n = 1), trimethoprim sulphonamide (TS) (n = 1), and clindamycin (n = 1). GCCs were administered in 9/30 (median time 2 days 0.5 mg/kg SID in 2/9 and BID in 7/9), NSAIDs were administered in 5/30 (median time 3 days) and in 4/30 antiemetic (maropitant citrate) was used.

Haematology was performed in all dogs. Mild neutrophilia (median 18.928 k/µL [5.05–16.76 k/μL]) was present in 3/30 (10%) of patients. Serum biochemistry results were unremarkable in all patients. Other ancillary studies included abdominal ultrasound on 12/30, urinalysis on 10/30, and thyroid hormones (total T4 and TSH) on 5/30, with uneventful results.

Bacteriological cultures performed in urine samples in four dogs and in blood in one dog were all negative. Middle ear sampling cultures were performed in 3/30, and the isolates included *Streptococcus viridians* (n = 1), Coagulase-Negative Staphylococci (n = 1), and *Escherichia coli* (n = 1).

### 3.3. Neurological Examination

Regarding the onset, signs were chronic in 11/30 (36.7%), acute in 10/30 (33.3%), and subacute in 9/30 (30%) dogs. Mental status was abnormal in 10/30 (33.3%) dogs and included disorientation (4/30), obtundation (5/30), and stupor (1/30), with all of them included in MRI groups B and C.

Vestibular ataxia was present in 29/30 dogs (not evaluated in the stuporous dog). It was characterized as mild in 15/29 (51.7%), moderate in 8/29 (27.6%), and severe in 6/29 (20.7%) dogs. The distribution of the ataxia in the different MRI groups is shown in Figure 2.

Head tilt (towards the side of the affected tympanic cavity if unilateral and to the most severely affected side if bilateral) was present in all dogs. Positional strabismus was observed in 15/30 (50%) dogs and spontaneous nystagmus in 5/30 (16.7%) cases (4/5 horizontal and 1/5 rotatory). Facial paralysis was observed in 11/30 (36.7%) dogs and was ipsilateral to the vestibular signs in all cases. Proprioceptive deficits ipsilateral to the head tilt were noticed in 10/30 (33.3%) cases.

In the studied group, for 26/30 (86.7%) dogs, the neuro-anatomical localization was defined as vestibular dysfunction (central in 6/30 patients and peripheral in 20/30). The remaining 4/30 dogs (13.3%) presented vestibular signs together with signs of multifocal brain disease. The group of 20/30 (66.7%) dogs in which intracranial extension was not recognized on neurological examination comprised all the six dogs from group A, 12/20 of MRI group B, and 2/20 of group C, as shown in Figure 3.

### 3.4. Magnetic Resonance Imaging (MRI) Features

The presence of soft tissue material, T2W hyperintense with heterogenous T1W post-gadolinium contrast enhancement, was found partially or completely filling either one tympanic cavity (16/30, 53.3%; 8/16 right and 8/16 left) or both (14/30, 46.7%). Other findings included unilateral inflammatory changes in para-aural soft tissues with moderate to marked contrast enhancement (26/30, 86.7%), meningeal thickening and contrast enhancement (24/30, 80%), and brainstem intra-axial signal intensity changes with an extra-axial collection consistent with empyema (7/30, 23.3%). Signs of mass effect (5/30, 16.7%; 1/5 with foramen magnum herniation), and facial and vestibulocochlear nerve thickening with contrast enhancement ipsilateral to other changes (2/30, 6.7%) were also noted. According to the MRI findings, 6/30 (20%) dogs were included in group A, 17/30 (56.7%) in group B, and 7/30 (23.3%) in group C. The classification was coincident for the two observers.

### 3.5. Cerebrospinal Fluid (CSF) Analysis

CSF analysis was performed in 29/30 dogs and revealed abnormalities in all of them. Pleocytosis with >75% of neutrophils, with variable degrees of degenerative changes, was observed in all the samples with a median TNCC of 4707 cells/µL (range 7–50,000 cells/μL). The median protein concentration was 96.5 mg/dL (range 25–400 mg/dL) and was higher in MRI group C (median of 187.5 mg/dL). Intracellular bacteria were not observed. CSF cultures (for bacteria and fungus) were performed in 16/29, being positive for bacteria in 3/16 (19%) with isolates including *Moraxella* spp. (n = 1), *Enterobacter cloacae* (n = 1), and *Staphylococcus Pseudointermedius* (n = 1). The case, in which CSF was not retrieved due to signs of cerebellar herniation in the MRI, was considered to have intracranial extension of the infection and was included in group C based on the MRI findings.

### 3.6. Treatment

Two dogs (a French bulldog and an English bulldog) were humanely euthanized after diagnosis due to the severity of the clinical signs. Both dogs had evidence of intracranial extension in MRI and were classified in group C; the English bulldog was the case presenting signs of foramen magnum herniation.

The remaining 28 dogs were all medically treated after diagnosis with broad-spectrum antibiotics. If the dogs were already on an antimicrobial treatment, the same regimen was maintained and combined with other antimicrobials to broaden the spectrum, considering the presence of an active infection despite previous treatment. The treatment of the six dogs with positive cultures either in CSF (3/6) or from ear material (3/6) was adjusted to the sensitivity results. In these cases, enrofloxacin was discontinued in 2/6 due to resistance and was changed to ACA, and in the remaining 4/6 cases, metronidazole was withdrawn and pre-existing antibiotics were continued with ACA (2/4), cephalexin (1/4), and enrofloxacin (1/4).

Seven out of thirty patients were hospitalized for 3–7 days to receive the treatment (five from group B and two from group C); of those, 3/7 received a combination of three antibiotics: TS (15 mg/kg/BID, IV), metronidazole (15 mg/kg/BID, IV), and enrofloxacin (5 mg/kg/SID, subcutaneous [SQ]); 4/7 received a combination of two antibiotics: 2/7 cefazolin (20 mg/kg/BID, IV) and enrofloxacin (5 mg/kg/SID, SQ), 1/7 marbofloxacin (2 mg/kg/SID, SQ) and ACA (20 mg/kg/BID, IV), and 1/7 enrofloxacin (5 mg/kg/SID, SQ) and TS (15 mg/kg/BID, IV). Prednisolone (0.5 mg/kg/BID, IV, initially 2 days and then tapering) was administered in 5/7 dogs and 2/7 received meloxicam (0.1 mg/kg/SID, SQ). Methadone (0.2 mg/kg/4 h, IV/IM) was administered for 24 h in 3/7 followed by buprenorphine (10 ug/kg/TID, IV) until the dogs were considered to not be in pain. Maropitant (1 mg/kg/SID, SQ) was used in 6/7. Artificial tears were administered in all of them.

The remaining 21 dogs were treated at home as follows: 11/21 received a combination of three antibiotics: TS (15 mg/kg/BID, PO), metronidazole (15 mg/kg/BID, PO), and enrofloxacin (5 mg/kg/SID, PO); 7/21 were treated with two antibiotics: 3/21 cephalexin (20 mg/kg/BID, PO) and metronidazole (15 mg/kg/BID, PO), 1/21 marbofloxacin (2 mg/kg/SID PO) and ACA (22 mg/kg/BID, PO), 1/21 ACA (20 mg/kg/BID, PO) and metronidazole (15 mg/kg/BID, PO), 1/21 enrofloxacin (5 mg/kg/SID, PO) and metronidazole (15 mg/kg/BID, PO), 1/21 ACA (20 mg/kg/BID, PO) and TS (15 mg/kg/BID, PO). The remaining 3/21 dogs were treated with one antibiotic: 1/21 TS (15 mg/kg/BID, PO), 1/21 cephalexin (20 mg/kg/BID, PO), and 1/21 enrofloxacin (5 mg/kg/SID, PO). Prednisolone (0.5 mg/kg/BID, PO, initially 2 days and then tapering) was administered in 12/21 dogs, and NSAIDs in 9/21 (7/9 meloxicam 0.1 mg/kg/SID, PO and 2/9 robenacoxib 1 mg/kg/SID, PO). In addition, 3/21 received tramadol (3 mg/kg/TID, PO, 3–5 days) and 2/21 paracetamol (10 mg/kg/8 h, PO, 3 days). Maropitant (1 mg/kg/SID, PO) was administered on 4/21 for 3 days. Artificial tears were administered in all of them.

The median antimicrobial course length in all dogs was 9.5 weeks (range 6–16 weeks). Prednisolone was administered for a median of 12 days (range 7–84 days) and NSAIDs for 8 days (range 3–15 days).

### 3.7. Follow-Up Magnetic Resonance Imaging (MRI) and Cerebrospinal Fluid (CSF)

Follow-up MRI studies (5/30) were performed from 5 weeks to 6 months after treatment initiation and included three dogs from group B (3/17 dogs, 17%) and two dogs from group C (2/7 dogs, 28%). Radiological improvement was noticed in all dogs. No signs of the previously observed epidural material or soft tissue enhancement were detected. In four dogs, a decrease (up to 50%) in the occupation of the tympanic cavities was observed, even though some degree of content was still present (Figure 4).

Follow-up CSF was performed in 3/5 dogs that underwent follow-up MRI, and in all of them, the results were within the normal range.

### 3.8. Follow-Up and Outcome

Long-term follow-up (range from 12 weeks to 24 months, average 16.8 weeks) was documented in all the treated dogs (28/30). Complete recovery was achieved in 8/28 (28.6%); the distribution in the MRI groups is shown in Figure 5. The remaining 20/28 dogs improved but showed mild persistent neurological deficits which included head tilt in 9/20 (45%), residual facial paralysis in 5/20 (25%), corneal ulcers in 4/20 (20%), mild vestibular ataxia in 1/20 (5%), and hearing impairment (perception of the owner) in 1/20 (5%) dogs.

Dogs with brainstem parenchymal involvement had a longer median recovery time (8 days for group A, 11 days for group B, and 21 days for group C) and 71.4% of dogs with sequels belonged to groups B and C. The two dogs that were humanely euthanized belonged to group C.

### 3.9. Statistical Analysis

No statistically significant differences between the MRI groups were obtained regarding either the neuro-anatomical localization or the presence of persistent neurological deficits. The results of OLS did not identify any statistically significant coefficients.

Descriptive results are shown in Appendix A and summarized in Figure 2, Figure 3, and Figure 5. In general, as we increased the MRI classification (A, B, C), dogs tended to be older, had more incidence of chronic or recurrent otitis, and had a higher degree of ataxia. Furthermore, dogs in groups B and C were older than 7 years with a median age of 9.7 years in group C, all dogs in group C had a history of chronic otitis, and in group C, ataxia at presentation was severe in 66.7% of dogs.

## 4. Discussion

Intracranial extension of OMI is a severe condition requiring urgent therapeutic interventions [1]. In a recent study about bacterial meningitis or meningoencephalitis in dogs, an otogenic source of infection was diagnosed in 63% of the cases [3].

The pathogenesis of OMI is likely to be multifactorial [6]. Bacterial virulence, anatomical defects, and altered host immunity may influence whether OMI extends into the cranial cavity [4]. In the study presented, 29/30 were brachycephalic dogs; these breeds are predisposed to middle ear effusion which is characterized by the presence of fluid in the bullae [3,15]. The presence of a middle ear effusion might increase susceptibility to secondary infections [16], similar to “glue ear” in children, occurring as a result of an inability of the Eustachian tube to drain away the fluid, increasing the likelihood of intracranial extension of OMI [17]. Whether the brachycephalic conformation promotes intracranial extension of the infection remains to be studied [3]. Most canine patients with OM also have chronic OE with pathologic changes in the ear canal that cause stenosis, making a visual examination of the TM impossible [17]. In the study presented, 10/30 (33%) dogs had signs of OE at the time of admission and 14/30 (46%) had a history of chronic OE. It has been reported that secondary OM occurs in approximately 16% of acute OE cases in dogs and as many as 50–80% of chronic OE cases [17], because of the close anatomical relationship between the external, middle, and inner ear structures [18]. The fact that OM is present in more than half of canine patients with chronic OE should alert veterinarians about its possible presence [17]. In our study, a history of chronic otitis was apparently correlated with the MRI group; all dogs in group C (with brainstem lesions) had a history of chronic or recurrent otitis.

Regarding the neurological examination, decreased mentation, proprioceptive deficits, and cranial nerve deficits can help to confirm the presence of CNS involvement, but the absence of these findings does not exclude it. Recent studies have demonstrated the limitations of using neurological examination findings to reliably differentiate between CVS and PVS in dogs [11,17]. The most frequent etiology in these cases with intracranial lesions initially diagnosed as PVS was inflammation (mainly meningoencephalitis secondary to otitis inter-media or meningoencephalitis of unknown origin, MUO), and it was associated with a rapid deterioration of neurological signs [19]. In the present study, dogs were initially considered as PVS in 20/30 (66.7%), CVS in 6/30 (20%), and multifocal in 4/30 (13.3%); therefore, presumed neuro-anatomical localization was accurate in 33.3% of cases (multifocal + CVS) which is lower compared with other studies [18]. Bongartz et al. (2020) determined that the neurological examination was efficient at identifying lesions in the CVS (98.4%) and less efficient at localizing lesions in the PVS (77.4%) [19]. We also have to consider the possibility of inaccuracy in determining if there was central involvement of the vestibular system due to subtle and therefore not recognized signs and/or to compensation in more chronic cases. As the diagnosis was reached shortly after the clinical neurological examination, the assumption of progressive clinical signs is unlikely.

MRI is the preferable imaging modality to study the ear and the CNS, facilitating the detection of a possible intracranial extension of the infection [12,13] and the most complete information to the owner. In our study, high-resolution 3D T1W-post-contrast sequences were included in some cases to improve the evaluation of all the inflammatory changes in OMI, in an attempt to better correlate them to the vestibular signs, and to potential meningeal/parenchymal lesions. In addition, in more severe cases, follow-up MRI can be useful to monitor the response to treatment. Nevertheless, in our study, 6/30 (20%) of cases did not show signs of intracranial extension of the OMI in the MRI, and intracranial extension from OMI cannot be excluded on MRI alone similarly to other studies [2]. Besides the inherent limitations of the MRI, the lack of contrast enhancement in spite of an inflammatory CSF may reflect that the degree of disruption of the blood-meningeal barrier is not enough to allow for the visualization of the enhancement [20].

When safe to collect, CSF analysis should be performed to support the diagnosis, but it must be considered that normal CSF does not rule out the possibility of bacterial infection [10]. As might be expected, in our study, the protein level was much higher in dogs with parenchymal lesions in MRI (group C). Cytopathological examination may be diagnostic if bacterial organisms are seen. In addition, the culture of infective organisms in CSF was achieved in three dogs. Moreover, pre-admission administration of antimicrobials may have affected the results of at least four CSF cultures (dogs with negative results and antibiotic treatment previous to admission), but it must be considered that CSF is reported to be often culture-negative [10]. In the case of negative cultures, tests such as 16S rRNA gene PCR/sequencing of CSF have been proposed to improve the identification of bacteria [21,22].

Middle ear culture was positive in all three dogs. Organisms that are typically identified in veterinary patients with OMI include *Staphylococcus intermedius*, *Streptococcus* spp., *Escherichia coli*, *Pseudomonas aeruginosa*, and *Proteus mirabilis*, similar to our results [1,2,17,23,24]. In human patients with intracranial sepsis, mixed infections are common, and anaerobic organisms are a frequent finding. The presence of anaerobic infection in veterinary patients with intracranial sepsis has also been described [10].

The management of otogenic meningoencephalitis needs urgent therapy, but the optimal antibiotic and duration of treatment remain unclear [3]. The chosen antibiotics should ideally be bactericidal and should achieve therapeutic concentrations within the CSF, regardless of the status of the blood-brain barrier [10]. In people, pending sensitivity results, the use of either penicillin or cephalosporins are common first-line agents for bacterial meningitis or meningoencephalitis [3]. The medical treatment used in this case series for dogs when the causative agent was not identified was a combination of antimicrobials to cover Gram-positive, Gram-negative, and anaerobes; the drug choices and duration varied depending on clinical severity and response. In some cases, it is recommended to remove necrotic and obstructing tissues and to promote drainage from the middle/inner ear as well as from the site of the intracranial extension [3,10]. All the cases of the present study were treated conservatively. The reasons for declining surgical treatment included a good short-term response to medical treatment and the risks of the surgical procedure; therefore, we cannot compare the outcomes between the two treatment options. In people, there is also a lack of available guidelines to describe the optimal management. A recent study concluded that the primary treatment of otogenic meningitis relies on broad-spectrum antibiotic therapy, which may be adjusted later according to bacterial cultures if available, with surgery employed in the event of complications and when initial treatment is not effective within 48 h [25].

Regarding the prognosis in dogs, it has been suggested that the onset of disease may affect the outcome, with those presenting with acute (<24 h) or subacute (1–7 days) signs more likely to have a suppurative OM/OI and a poorer outcome [4]. In our study, chronicity and age were associated with the MRI group; all dogs in group C had a history of chronic otitis and were significantly older (median age 9.7 years). Bacterial meningitis initiates a cascade, resulting in inflammation of the subarachnoid space, vasculitis, cerebral edema, and injury to cortical and subcortical brain structures. Ventriculitis occurs most commonly as a late and fatal complication of meningitis. The purulent exudate that is produced, impairs the CSF flow and resorption, resulting in the blockade of outflow tracts and hydrocephalus in some cases [4]. In people, neurologic sequelae after meningoencephalitis secondary to acute OMI are thought to develop as a result of the effects of inflammatory mediators and cytokines. As a result, glucocorticoids are routinely administered without evidence of interference with antibiotic treatment [1]. In the present study, prednisolone was administered in 17/30 dogs (median 12 days) and NSAIDs (median 8 days) in 11/30, with no statistical difference between the MRI group. In humans, the mortality rate of otogenic meningoencephalitis remains high, ranging from 5–14% in adults [4] and up to 10–41% [19] in children. In this study, two dogs (6.7%) were euthanized at the time of the diagnosis due to the severity of the lesions and clinical signs. We have to consider that we cannot exclude a possible intracranial process concurrent with the OMI, such as lymphoma, for the case with cerebellar herniation that was euthanized without CSF analysis, but this seems improbable considering the clinical data and MRI findings.

On the other hand, all the 28 remaining patients showed improvement with no signs of recurrence at the time of long-term follow-up. The definition of short-term and long-term periods varies between studies. In our study, 6 weeks was the minimum treatment length; therefore, it was chosen as the short-term period, and a long-term outcome of a minimum of 12 weeks was decided to ensure the clinical signs were stabilized. Complete recovery was achieved in 28.6% of dogs with the majority of them belonging to group A. When observed, persistent deficits were considered to be mild with a similar percentage of dogs from groups B and C. The most common residual neurological deficits were head tilt and corneal ulceration related to facial paralysis. In this context, it must be noted that brachycephalic breeds are predisposed to corneal ulcers [26], which may complicate management. The median recovery time was longer in group C and shorter in group A, which may indicate that the dogs with more severe status may need a longer recovery time. In conclusion, dogs with brainstem parenchymal involvement (group C) had a longer recovery, and groups B and C dogs were more likely to have persistent neurological deficits. The proposed MRI classification may help to establish a prognosis in the clinical setting. Nevertheless, although coincident for the observers in our study, further studies with inter-rater agreement analysis would be advisable.

The study is subject to several limitations, primarily the relatively small sample size and the unequal distribution of cases across the MRI-based classification categories, which hinder the ability to obtain statistically significant results. Additional limitations include the retrospective design, the lack of availability of high-resolution 3D sequences in all the studies, the absence of surgically treated cases, and the variability in clinical approaches, which result in a lack of standardization in treatment protocols. The interpretation of long-term follow-up data should be cautiously approached, as it was collected through telephone interviews with RVS/owners. Furthermore, the number of follow-up MRIs is limited, which is noteworthy given that such studies are typically constrained in clinical practice due to financial and anesthesia-related considerations.

## 5. Conclusions

The neurological examination by itself is not always accurate in identifying intracranial extension of ear infection in dogs, and otogenic meningoencephalitis should be considered even in patients with PVS. MRI and CSF should be advised in certain cases to allow for diagnosis. Dogs belonging to brachycephalic breeds, older, with chronic or recurrent otitis, and with severe ataxia, were more likely to have intracranial extension of the OMI. Although it bears grave consequences for the patient if not recognized and addressed appropriately, excellent outcomes can be achieved with medical management. Further studies which include comprehensive ancillary diagnostics are needed to identify the most common pathogens and to elaborate treatment guidelines.

## Figures and Tables

**Figure 1 vetsci-12-00456-f001:**
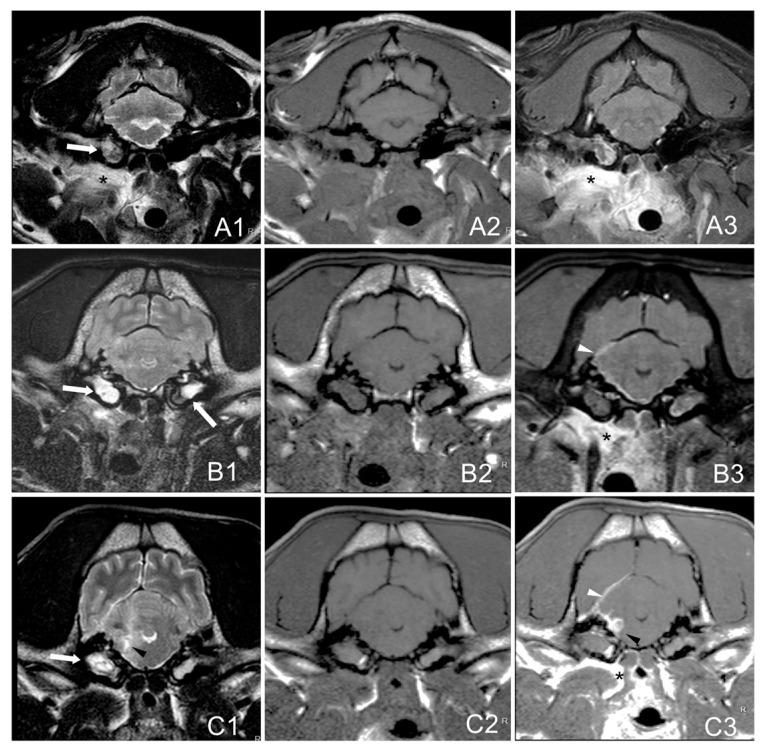
Magnetic resonance images (MRIs) of the head on a transverse plane at the level of the tympanic bullae of three dogs with otitis inter-media showing the proposal MRI grading system. T2W: (**A1**,**B1**,**C1**). T1W: (**A2**,**B2**,**C2**). Post-contrast-T1W: (**A3**,**B3**,**C3**). Group **A**. French bulldog, 4 years (**A1**–**A3**): bulla occupation (white arrow) and changes in surrounding soft tissues (asterisk) without apparent radiological encephalon extension. Group **B.** French bulldog, 7 years (**B1**–**B3**): bullae occupation (white arrows), changes in surrounding soft tissues (asterisk) and imaging of meningeal enhancement (white arrowhead). Group **C**. French bulldog, 9 years (**C1**–**C3**): bullae occupation (white arrows), changes in surrounding tissues (asterisk), meningeal thickening (white arrowhead), and brainstem space-occupying accumulation of epidural material corresponding to the formation of an abscess or empyema T2W-hyperintense with peripheral heterogenous contrast enhancement (black arrowheads).

**Figure 2 vetsci-12-00456-f002:**
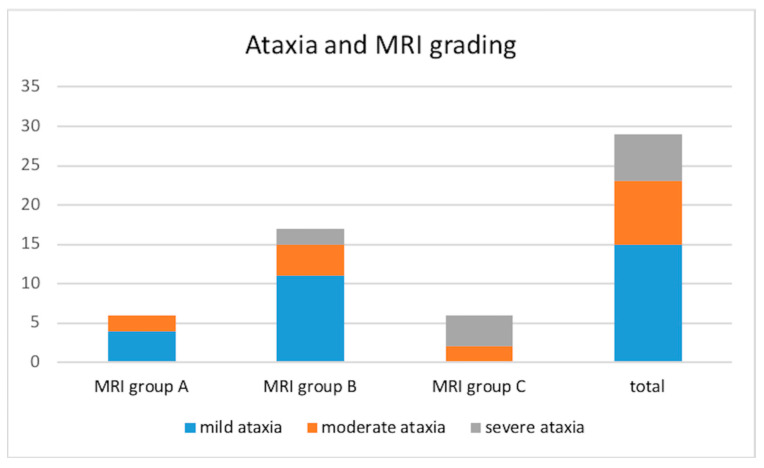
Frequencies of the different grades of ataxia within each MRI group in 29 dogs with otogenic meningoencephalitis or meningitis. Group A: bulla/bullae occupation and changes in surrounding soft tissues without evidence of extension to the adjacent meninges or neuroparenchyma; Group B: bulla/bullae occupation, changes in surrounding soft tissues and imaging of meningeal post-contrast enhancement; Group C: bulla/bullae occupation, changes in surrounding tissues, meningeal thickening and/or enhancement and brainstem lesion with or without the formation of an empyema.

**Figure 3 vetsci-12-00456-f003:**
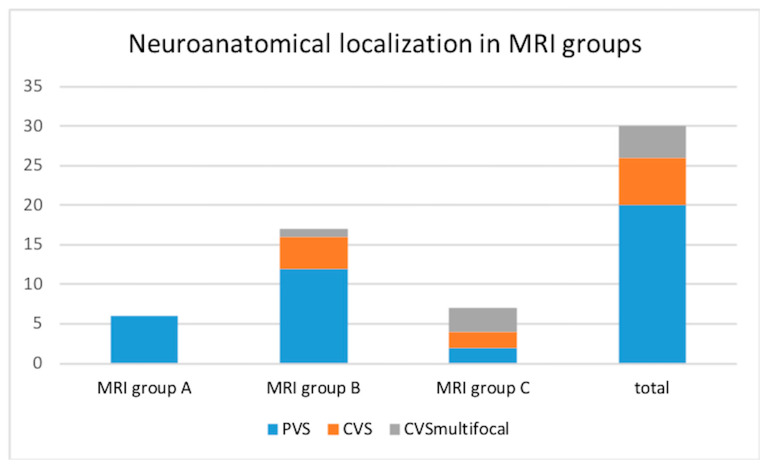
Relationship between neuro-anatomical localization and MRI grading in 30 dogs with otogenic meningoencephalitis or meningitis. Group A: bulla/bullae occupation and changes in surrounding soft tissues without evidence of extension to the adjacent meninges or neuroparenchyma; Group B: bulla/bullae occupation, changes in surrounding soft tissues and imaging of meningeal post-contrast enhancement; Group C: bulla/bullae occupation, changes in surrounding tissues, meningeal thickening and/or enhancement and brainstem lesion with or without the formation of an empyema. PVS: peripheral vestibular syndrome; CVS: central vestibular syndrome.

**Figure 4 vetsci-12-00456-f004:**
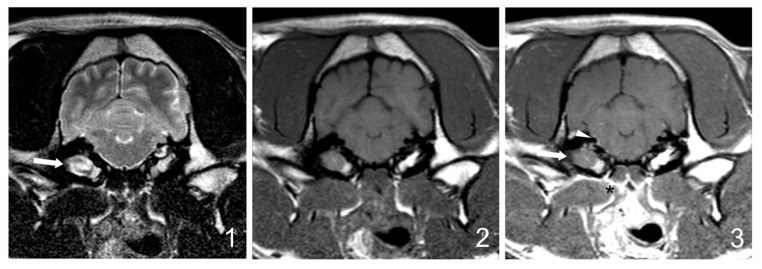
Follow-up MRI images (1: transverse T2W, 2: transverse T1W, 3: transverse T1W after gadolinium administration at the level of the tympanic bullae) of the 9-year-old French bulldog from Group C shown in Figure 1, after medical treatment (enrofloxacin and TS for 12 weeks). The brainstem occupation (black arrowheads in Figure 1 [C1 and C3]) is not visible, and the degree of contrast enhancement in surrounding soft tissues (black asterisk) and meninges (white arrowhead) is mild and focal compared to the previous MRI. No significant changes in the bullae occupation are observed (white arrow).

**Figure 5 vetsci-12-00456-f005:**
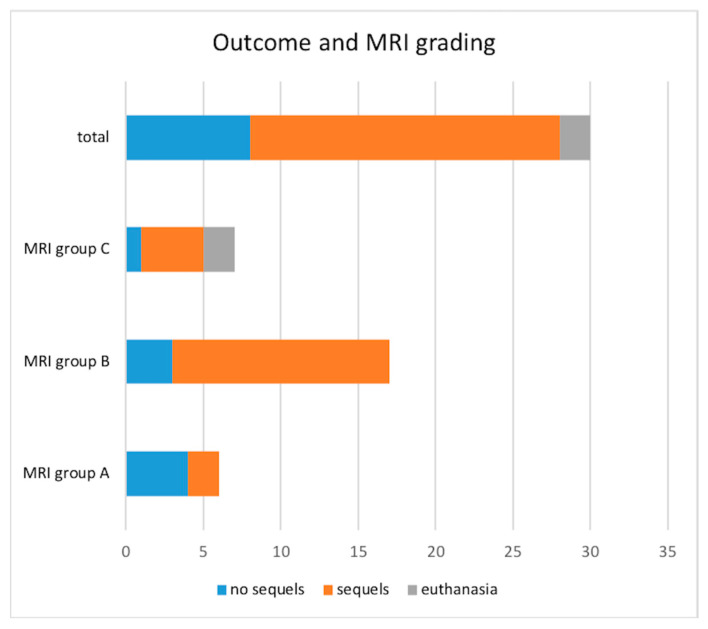
Outcome of 30 dogs with otogenic meningoencephalitis or meningitis. Group A: bulla/bullae occupation and changes in surrounding soft tissues without evidence of extension to the adjacent meninges or neuroparenchyma; Group B: bulla/bullae occupation, changes in surrounding soft tissues and imaging of meningeal post-contrast enhancement;, Group C: bulla/bullae occupation, changes in surrounding tissues, meningeal thickening and/or enhancement and brainstem lesion with or without the formation of an empyema.

## Data Availability

The data used and analyzed in the current study are available from the corresponding author upon reasonable request.

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
