# Peer review of "Otogenic Meningitis or Meningoencephalitis in 30 Dogs: Association Between Neurological Signs, Magnetic Resonance Imaging Findings, and Outcome"

_vetsci, 2025, doi:10.3390/vetsci12050456_

Round 1
Reviewer 1 Report
Comments and Suggestions for Authors
Unfortunately, due to major limitations, I cannot recommend this MS for publication in the current form. More work might be needed to fill important gaps:
A key information—assessing whether the neurological examination effectively identified lesions in the CVS versus PVS— fails to come through, possibly due to inconsistencies in the presentation of the results.
There are ambiguous statements in the discussion regarding certain results, which are described as statistically significant or “at risk” despite the absence of supporting statistical data.
Reference to previous studies fails, on occasion, to deliver the information that has a bearing on the questions examined
I have provided some comments below of specific areas where I suggest editing or adding clarification.
Line 47: [..] “MRI also allows to obtain high resolution three-dimensional (3D) images”[..] the relevance of this specification in the context of the MS (otogenic meningitis) is unclear. While MRI has other applications, it is not evident why the 3D imaging function is particularly pertinent to the scope of the MS. Please specify or consider removing
Line 61: "Neutrophilic pleocytosis (>50%) with degenerate neutrophils in the cerebrospinal fluid (CSF) analysis was noted where performed. "cases with and without CSF analysis were included; however, radiological findings of otogenic meningitis with intracranial extension and other pathological processes, such as lymphoma, may overlap. Please ensure this is acknowledged in the discussion as a limitation
Line62 :[..] recorded short (from the diagnosis to 6 weeks) and long-term outcome (from 6 weeks to a minimum of 12 weeks following diagnosis). [..] what is the rationale for these time frames?
Line 68 :[..] results of other ancillary tests were also noted [..] Please specify which test or at least provide those as an addendum in supplementary files.
Line 72:[..] Dogs were classified as having a peripheral vestibular syndrome or a central vestibular syndrome (either alone or together with other intracranial signs) [..] Although differentiating central from peripheral vestibular syndrome is explained elsewhere, it should be useful to report here a list of the clinical signs, suggestive of a central location. This would avoid any confusion in data interpretation later. (i.e. what is Author’s conclusion on dogs with vestibular signs and normal proprioception but lethargic? ). In addition to this, the inclusion criteria here reported do not match the description of results (Line 226: [..] 4/30 dogs (13.3%) presented signs of multifocal brain disease [..] did these dogs presented with multifocal brain disease with or without vestibular signs? Please re-write the two paragraphs and revise for clarity.
Line 84 :[..] High-resolution 3D T1W sequences were included in some studies better to evaluate the presence of facial or vestibulocochlear nerve changes. [..] clearly, due to the retrospective nature of the study availability of the sequence of interest varies among this cohort, however, it may be important to add it as a limitation in the discussion.
Line 100:[..] perilesional edema, [..] it may be stressed that perilesional oedema per se does not represent a sign of increased intracranial pressure. Please change
Line 369:[..] In our study, a history of chronic otitis was significantly correlated with the MRI group [..] within the MS it is not explicitly demonstrated that the correlation between a history of chronic otitis and the MRI group is statistically significant. Without the reported results of a statistical test, we cannot definitively say that the correlation was statistically significant.– one thing is providing descriptive data reporting that all the (7) dogs in group C had chronic history of signs, another is attributing statistical significance to it - please remove this confusing statement
Line 226 to 230 :[..] The group of 20/30 (66.7%) dogs in which intracranial extension was not recognized on neurological examination comprised 12/20 of MRI group B, 6/20 of MRI group A, and 2/20 of group C. Considering the total number of dogs, 40% of cases belonged to Group B, 20% to Group 229 A, and 6.7% to Group C. [..]It is very challenging to understand how many dogs were localised correctly in the group A (as this includes dogs with radiological finding of OMI without intracranial extension, and all 6 dogs are localised PVS according to fig.3, the statement “intracranial extension was not recognized on neurological examination in 6/20 of MRI group A “does not seem to be correct. Please re-write the entire paragraph and revise for clarity.
Line 334:[..] In general, as we increased the MRI classification (A, B, C), [..] please rewrite, review for clarity/English
Line380 :[..] therefore presumed neuroanatomical localization was accurate in 36.6% [..] it is not clear how this result is obtained and overall, how accuracy (%) of the neuroanatomical localization was determined. Please consider rewriting the entire paragraph linking this with the data provided/ammended in the result section
Line 387:[..] We found that inaccuracy in neuroanatomical localization at admission was higher in group A (83%), which had less severe lesions on MRI. [..] this information is not present in the result section
Line 393:[..] efforts to reach a definitive diagnosis should be pursued in dogs with signs of PVS if they do not respond to treatment. [..] it is generally a good advice, but the data provided in this study do not support this statement
Line427 : By introducing and reviewing items of previous research/reports it should select information in closer association with the aim of the manuscript, overall, in the discussion this is not always the case.
Line470 :This final paragraph fails to align with the aims set out in the manuscript
Fig. 2: The title of Figure 2 "Relationship between ataxia and MRI grading" is not entirely appropriate, as the figure simply shows histogram of the frequencies of the different grades of ataxia within each MRI group, rather than a direct relationship between ataxia and MRI grading. Please change.
Fig. 3 & 5: The graphs in Fig 3 and 5 do not provide much additional information beyond what is already described in the text. The text already details the differences in neuroanatomical localization and the presence of persistent neurological deficits across the different MRI groups. Without any normalization or statistical analysis of the data, the graphs simply present the raw frequencies, which do not clearly highlight the differences between groups. Consider removing
Author Response
Dear reviewer,
Thank you for your detailed and useful comments. Thorough changes have been made in the text according to them and are marked in red to facilitate identification. You may find the answers to your specific comments below your comments (in blue).
Unfortunately, although differences between groups were found, they were not statistically significant. This is now clearly stated in the results and commented on the limitations. The concept “at risk” has been deleted.
Line 47: [..] “MRI also allows to obtain high resolution three-dimensional (3D) images”[..] the relevance of this specification in the context of the MS (otogenic meningitis) is unclear. While MRI has other applications, it is not evident why the 3D imaging function is particularly pertinent to the scope of the MS.
We understand the reviewer comment. Information about the use of high-resolution 3D T1W-postcontrast sequences was expanded in the conclusion.
[..] clearly, due to the retrospective nature of the study availability of the sequence of interest varies among this cohort, however, it may be important to add it as a limitation in the discussion.
This has been added to the limitations.
Line 61: "Neutrophilic pleocytosis (>50%) with degenerate neutrophils in the cerebrospinal fluid (CSF) analysis was noted where performed. "cases with and without CSF analysis were included; however, radiological findings of otogenic meningitis with intracranial extension and other pathological processes, such as lymphoma, may overlap. Please ensure this is acknowledged in the discussion as a limitation.
We thank the reviewer for the comment, but we have to consider the following:
- CSF analysis was performed in all cases except the one with signs of cerebellar herniation and showed pleocytosis with >75% neutrophils in all the samples.
- Antibiotic treatment resulted in improvement of the signs to total recovery or mild sequels in all the treated cases.
- The only two cases that were not treated but humanely euthanized following diagnosis had MRI features of group C: bulla/bullae occupation, changes in surrounding tissues, meningeal thickening and enhancement, and brainstem lesion. For those two cases, all clinical data supported the diagnosis of otogenic meningitis with intracranial extension, one of them supported by the results of CSF.
- We understand that the reviewer´s intention is to point out that for the case with cerebellar herniation that was euthanized, we must consider a possible intracranial process concurrent with the otitis. To follow the request, we added a comment in the discussion.
Line62 :[..] recorded short (from the diagnosis to 6 weeks) and long-term outcome (from 6 weeks to a minimum of 12 weeks following diagnosis). [..] what is the rationale for these time frames?
As the definition of short-term and long-term periods varies between studies, we chose what we believed best suited for our study. Six weeks was the minimum treatment length, so it was chosen as the short-term period. A long-term outcome of 12 weeks was decided to ensure the clinical signs were stabilized. We added a sentence regarding this in the discussion.
Line 68 :[..] results of other ancillary tests were also noted [..] Please specify which test or at least provide those as an addendum in supplementary files.
The sentence was changed to: “If performed, results of bacterial culture from urine, blood, CSF and/or middle ear samples were also noted.”
Line 72:[..] Dogs were classified as having a peripheral vestibular syndrome or a central vestibular syndrome (either alone or together with other intracranial signs) [..] Although differentiating central from peripheral vestibular syndrome is explained elsewhere, it should be useful to report here a list of the clinical signs, suggestive of a central location. This would avoid any confusion in data interpretation later. (i.e. what is Author’s conclusion on dogs with vestibular signs and normal proprioception but lethargic? ).
Thank you for the comment. As the reviewer points out the description of the signs in case of PVS and CVS is given before, in the introduction, and it was nor repeated later to avoid redundancy. According to the comment “abnormal mental status” was added to the description.
In addition to this, the inclusion criteria here reported do not match the description of results (Line 226: [..] 4/30 dogs (13.3%) presented signs of multifocal brain disease [..] did these dogs presented with multifocal brain disease with or without vestibular signs? Please re-write the two paragraphs and revise for clarity.
Thank you for the comment. In point 2.2 it is described: “Dogs were classified as having a PVS or a central vestibular syndrome (CVS) (either alone or together with other intracranial signs).” For better clarity this has been added again in the results.
Line 84 :[..] High-resolution 3D T1W sequences were included in some studies better to evaluate the presence of facial or vestibulocochlear nerve changes. [..] clearly, due to the retrospective nature of the study availability of the sequence of interest varies among this cohort, however, it may be important to add it as a limitation in the discussion.
This is now commented in the discussion and in the limitations.
Line 100:[..] perilesional edema, [..] it may be stressed that perilesional oedema per se does not represent a sign of increased intracranial pressure. Please change
We understand the reviewer´s concern. The description of the signs of elevated ICP in dogs is based on the literature and referenced (number 14 in reference list, Bittermann S, Lang J, Henke D, Howard J, Gorgas D. Magnetic resonance imaging signs of presumed elevated intracranial pressure in dogs. Vet J. 2014 Jul;201(1):101-8. doi: 10.1016/j.tvjl.2014.04.020. Epub 2014 May 6. PMID: 24888678.)
Line 369:[..] In our study, a history of chronic otitis was significantly correlated with the MRI group [..] within the MS it is not explicitly demonstrated that the correlation between a history of chronic otitis and the MRI group is statistically significant. Without the reported results of a statistical test, we cannot definitively say that the correlation was statistically significant.– one thing is providing descriptive data reporting that all the (7) dogs in group C had chronic history of signs, another is attributing statistical significance to it - please remove this confusing statement
We apologize for the misuse of the term, “significantly” was changed to “apparently”.
Line 226 to 230 :[..] The group of 20/30 (66.7%) dogs in which intracranial extension was not recognized on neurological examination comprised 12/20 of MRI group B, 6/20 of MRI group A, and 2/20 of group C0. Considering the total number of dogs, 40% of cases belonged to Group B, 20% to Group 229 A, and 6.7% to Group C. [..]It is very challenging to understand how many dogs were localised correctly in the group A (as this includes dogs with radiological finding of OMI without intracranial extension, and all 6 dogs are localised PVS according to fig.3, the statement “intracranial extension was not recognized on neurological examination in 6/20 of MRI group A “does not seem to be correct. Please re-write the entire paragraph and revise for clarity.
We apologize for the unclear description of the results. The text was changed for clarity.
Line 334:[..] In general, as we increased the MRI classification (A, B, C), [..] please rewrite, review for clarity/English
The text was changed accordingly.
Line380 :[..] therefore presumed neuroanatomical localization was accurate in 36.6% [..] it is not clear how this result is obtained and overall, how accuracy (%) of the neuroanatomical localization was determined. Please consider rewriting the entire paragraph linking this with the data provided/ammended in the result section.
We thank you the reviewer for pointing out the error, the correct number is 33.3%. This was corrected.
Line 387:[..] We found that inaccuracy in neuroanatomical localization at admission was higher in group A (83%), which had less severe lesions on MRI. [..] this information is not present in the result section
We apologize for the erroneous sentence, it was deleted.
Line 393:[..] efforts to reach a definitive diagnosis should be pursued in dogs with signs of PVS if they do not respond to treatment. [..] it is generally a good advice, but the data provided in this study do not support this statement
The sentence was deleted.
Line427 : By introducing and reviewing items of previous research/reports it should select information in closer association with the aim of the manuscript, overall, in the discussion this is not always the case.
A thorough review of the previous report’s citation was completed.
Line470 :This final paragraph fails to align with the aims set out in the manuscript
Changes were done accordingly.
Fig. 2: The title of Figure 2 "Relationship between ataxia and MRI grading" is not entirely appropriate, as the figure simply shows histogram of the frequencies of the different grades of ataxia within each MRI group, rather than a direct relationship between ataxia and MRI grading. Please change.
The title and legend were changed.
Fig. 3 & 5: The graphs in Fig 3 and 5 do not provide much additional information beyond what is already described in the text. The text already details the differences in neuroanatomical localization and the presence of persistent neurological deficits across the different MRI groups. Without any normalization or statistical analysis of the data, the graphs simply present the raw frequencies, which do not clearly highlight the differences between groups. Consider removing
We thank the reviewer for the comment. Information shown in the graphs has been removed from the text to avoid redundancy.
Reviewer 2 Report
Comments and Suggestions for Authors
Manuscript Title: Otogenic meningitis or meningoencephalitis in 30 dogs: association between neurological signs, MRI findings and outcome
Manuscript ID: [vetsci-3560689]
General Evaluation:
This manuscript addresses an important and relatively underreported condition in veterinary neurology—otogenic meningitis or meningoencephalitis (OME) in dogs. The retrospective analysis of 30 cases, with a proposed MRI-based grading system and outcome correlation, offers clinical relevance and novel contributions to the literature.
The study is generally well structured and well written. However, there are several points requiring clarification, refinement, and improvement to enhance the scientific rigor, reproducibility, and interpretability of the results.
I recommend major revisions before acceptance.
Major Comments:
- Statistical Analysis:
- The use of Ordinary Least Squares (OLS) regression is stated, but no statistical outputs (e.g., regression coefficients, p-values, or confidence intervals) are presented.
- Please include a results table summarizing the statistical findings. Additionally, consider whether OLS is the most appropriate model, given the ordinal nature of several variables (e.g., MRI group, severity of ataxia).
- Validation of MRI Grading System:
- The proposed classification (Groups A–C) is conceptually helpful but lacks validation or inter-rater agreement analysis.
- Consider discussing potential reproducibility and utility in clinical settings. Has the grading been reviewed independently by multiple observers?
- Treatment Heterogeneity and Impact:
- Antibiotic and adjunctive treatments varied widely between patients. Yet, the manuscript does not explore whether this influenced the recovery duration or presence of neurological sequelae.
- Consider subgroup comparisons or multivariable models, or at minimum acknowledge this limitation more explicitly in the discussion.
- MRI–CSF Discordance:
- In 6 cases, CSF abnormalities were noted without evidence of intracranial involvement on MRI. This discordance requires a more detailed discussion—whether it reflects MRI sensitivity limits or other pathophysiologic mechanisms.
- Follow-up Assessment Bias:
- Long-term outcomes were collected primarily through telephone interviews, which may be subject to recall and subjectivity bias. This should be clearly stated in the limitations.
Minor Comments and Stylistic Suggestions:
- Improve clarity of objectives (Line 49–52):
Suggested revision:
“This study aimed to determine the diagnostic accuracy of neurological examination in identifying intracranial extension in canine otitis media/interna, and to evaluate associations between MRI findings, clinical variables, and patient outcomes.”
- MRI Grading Descriptions:
Consider presenting Group A–C definitions in a table or bullet list in the Methods section for clarity.
- Terminology Consistency:
Ensure all abbreviations (e.g., OE, OMI, PVS, CVS) are defined at first mention in both abstract and main text, and repeated in table/figure legends.
- Tables and Figures:
- Table 1 is dense; consider simplifying it in the main text and placing full details in supplementary material.
- Figure captions are sometimes lengthy or redundant; streamline where possible.
- CSF Discussion:
Culture-negative CSF is briefly acknowledged. Consider referencing alternative diagnostics (e.g., PCR, 16S rRNA gene sequencing) as emerging options in future studies.
- Grammar and Style:
A few minor revisions are suggested:
- Line 14–15: “Further studies including full ancillary tests…” → “...comprehensive ancillary diagnostics...”
- Line 66: Rephrase for clarity: “...results from urine, blood, CSF, or middle ear cultures...”
- Limitations Section:
Consider explicitly listing the study’s main limitations in a short paragraph at the end of the discussion for transparency: retrospective design, non-standardized treatments, small number of follow-up MRIs, and outcome assessment bias.
- References:
Ensure consistency in formatting (e.g., italicization of journal names, punctuation). Minor inconsistencies are present.
Author Response
Dear reviewer,
Thank you for your detailed and useful comments. Thorough changes have been made in the text according to them and are marked in red to facilitate identification. You may find the answers to your specific comments below your comments (blue).
- Statistical Analysis:
- The use of Ordinary Least Squares (OLS) regression is stated, but no statistical outputs (e.g., regression coefficients, p-values, or confidence intervals) are presented.
- Please include a results table summarizing the statistical findings. Additionally, consider whether OLS is the most appropriate model, given the ordinal nature of several variables (e.g., MRI group, severity of ataxia).
We apologize for the missing information, which now is expanded in the text. In the case of MRI group, the variable was converted into a binary variable, to compare intracranial versus extracranial, where those belonging to group A were assigned a value of 1, and those in B or C were 0. Ataxia was codified as a categorical variable. When running the regression each category was considered as a separate dummy variable.
Unfortunately, the statistical analysis did not show significant results, therefore we considered it unworthy to add them to the text. If the reviewer considers that the information is essential, we can provide the results in the supplementary material.
- Validation of MRI Grading System:
- The proposed classification (Groups A–C) is conceptually helpful but lacks validation or inter-rater agreement analysis.
- Consider discussing potential reproducibility and utility in clinical settings. Has the grading been reviewed independently by multiple observers?
Thank you for the comments. All imaging studies were reviewed for diagnostic accuracy by two independent observers, and the classification was coincident. Information has been expanded in the results, and a comment on the utility, and the lack of evaluation of interrater agreement has been included in the discussion.
- Treatment Heterogeneity and Impact:
- Antibiotic and adjunctive treatments varied widely between patients. Yet, the manuscript does not explore whether this influenced the recovery duration or presence of neurological sequelae.
- Consider subgroup comparisons or multivariable models, or at minimum acknowledge this limitation more explicitly in the discussion.
Thank you for the comments. We do agree variability in managing is a limitation of the study, therefore a comment was included in the previous version. To address the reviewer´s indication information on this issue has been expanded (see point 7).
- MRI–CSF Discordance:
- In 6 cases, CSF abnormalities were noted without evidence of intracranial involvement on MRI. This discordance requires a more detailed discussion—whether it reflects MRI sensitivity limits or other pathophysiologic mechanisms.
A comment has been added with a new reference in the conclusion.
- Follow-up Assessment Bias:
- Long-term outcomes were collected primarily through telephone interviews, which may be subject to recall and subjectivity bias. This should be clearly stated in the limitations.
This point was addressed (point 7)
Minor Comments and Stylistic Suggestions:
- Improve clarity of objectives (Line 49–52):
Suggested revision:
“This study aimed to determine the diagnostic accuracy of neurological examination in identifying intracranial extension in canine otitis media/interna, and to evaluate associations between MRI findings, clinical variables, and patient outcomes.”
Many thanks for your helpful suggestion, the text was changed accordingly.
- MRI Grading Descriptions:
Consider presenting Group A–C definitions in a table or bullet list in the Methods section for clarity.
Thank you for the suggestion. A bullet list has been added.
- Terminology Consistency:
Ensure all abbreviations (e.g., OE, OMI, PVS, CVS) are defined at first mention in both abstract and main text, and repeated in table/figure legends.
The text was reviewed and changed accordingly.
- Tables and Figures:
- Table 1 is dense; consider simplifying it in the main text and placing full details in supplementary material.
The table is now provided in supplementary material.
- Figure captions are sometimes lengthy or redundant; streamline where possible.
The text has been simplified when possible, and related to the figures.
- CSF Discussion:
Culture-negative CSF is briefly acknowledged. Consider referencing alternative diagnostics (e.g., PCR, 16S rRNA gene sequencing) as emerging options in future studies.
Thank you for the useful comment. The information was added to the text with the corresponding references.
- Grammar and Style:
A few minor revisions are suggested:
- Line 14–15: “Further studies including full ancillary tests…” → “...comprehensive ancillary diagnostics...” Changed
- Line 66: Rephrase for clarity: “...results from urine, blood, CSF, or middle ear cultures...”
Changed to: “Results of bacterial culture from urine, blood, CSF and/or middle ear samples were also noted.”
- Limitations Section:
Consider explicitly listing the study’s main limitations in a short paragraph at the end of the discussion for transparency: retrospective design, non-standardized treatments, small number of follow-up MRIs, and outcome assessment bias.
The paragraph addressing the limitations was expanded.
- References:
Ensure consistency in formatting (e.g., italicization of journal names, punctuation). Minor inconsistencies are present.
Thank you for the comment, it was reviewed and corrected.
Round 2
Reviewer 2 Report
Comments and Suggestions for Authors
Manuscript Title: Otogenic meningitis or meningoencephalitis in 30 dogs: association between neurological signs, MRI findings and outcome
Manuscript ID: vetsci-3560689
Recommendation: Minor Revisions (Accept after minor edits)
Overall Assessment:
The authors have thoroughly addressed the comments from the first round of review. The manuscript is now considerably improved in clarity, depth, and organization. The expanded methodological details, improved discussion, and incorporation of suggestions (e.g., MRI grading system, limitations, CSF testing alternatives, figure simplifications) have strengthened the study.
Most importantly, the statistical methods have been clarified, and while the results were non-significant, their explanation in the Discussion adds to transparency.
Remaining Minor Points for Final Acceptance:
1. Statistical Output in Supplementary Material:
Although the authors state that results were non-significant, it is still encouraged to include a simple supplementary table showing the OLS regression output. This provides transparency and may help future meta-analyses.
➤ Suggestion: A brief supplementary table titled “OLS Regression Results” can include: dependent variable, predictor, coefficient, standard error, and p-value.
2. Minor Language Corrections:
A few grammar or clarity edits are still recommended. Examples:
- Line 337: “No statistically significant differences between MRI groups was obtained…”
→ “...were obtained.”
- Line 451: “...euthanized without CSF analysis, but this seems not probable...”
→ “...but this is unlikely...” or “...but this seems improbable...”
3. Conclusion Section – Typo:
- Line 487: “...which include full ancillary tests...”
→ Should match the phrasing in the Simple Summary: “...comprehensive ancillary diagnostics...”
Strengths Retained in the Revised Version:
- Clear presentation of clinical relevance
- Excellent MRI image examples (Figures 1 & 4)
- Well-organized grading system (Groups A–C)
- Acknowledgment of diagnostic limitations (e.g., MRI–CSF discordance)
- Honest discussion of treatment heterogeneity
- Expanded and well-reasoned limitations section
Final Recommendation: Minor Revisions
With the addition of the statistical output in the supplementary material and minor language polishing, the manuscript is acceptable for publication.
Author Response
1. Statistical Output in Supplementary Material:
Although the authors state that results were non-significant, it is still encouraged to include a simple supplementary table showing the OLS regression output. This provides transparency and may help future meta-analyses.
➤ Suggestion: A brief supplementary table titled “OLS Regression Results” can include: dependent variable, predictor, coefficient, standard error, and p-value.
Thank you for the suggestion. Two tables showing OLS results are added in supplementary material.
2. Minor Language Corrections:
A few grammar or clarity edits are still recommended. Examples:
- Line 337: “No statistically significant differences between MRI groups was obtained…”
→ “...were obtained.”
- Line 451: “...euthanized without CSF analysis, but this seems not probable...”
→ “...but this is unlikely...” or “...but this seems improbable...”
3. Conclusion Section – Typo:
- Line 487: “...which include full ancillary tests...”
→ Should match the phrasing in the Simple Summary: “...comprehensive ancillary diagnostics...”
Answer: We apologize for the mistakes. These changes are marked in blue in the text.
